# Remote Sensing Object Detection Based on Fusion of Spatial and Channel Attention

Wenyun Sun
*School of Artificial Intelligence*
*Nanjing University of*
*Information Science and*
*Technology*
Nanjing, China
wenyunsun@nuist.edu.cn

Long Ji
*School of Computer Science*
*Nanjing University of*
*Information Science and*
*Technology*
Nanjing, China
202212490374@nuist.edu.cn

*Abstract*—Remote sensing object detection faces unique challenges due to objects' varied scales and orientations. To address these challenges, we propose the Spatial Channel Attention Fusion Module (SCAF-Module), designed to enhance detection accuracy by integrating multi-scale convolutions, adaptive rotated convolutions, and parallel spatial channel attention mechanisms. The experiments, conducted using the DOTA-v1.0 and HRSC2016 datasets, demonstrate the efficacy of the SCAF-Module. We achieved mean Average Precision (mAP) scores of 80.94% and 98.23% on these datasets, respectively. Comparative experiments reveal that the SCAF-Module surpasses several advanced models, including the baseline Oriented R-CNN. Additionally, ablation studies highlight the significance of the spatial and channel attention mechanisms and the impact of rotated convolutions on detection performance. The SCAF-Module presents a robust and adaptable framework for remote sensing object detection, offering significant improvements over existing methods. This work paves the way for further optimization and application of the module in other challenging remote sensing tasks.

*Keywords—Remote Sensing, Object Detection, Attention Mechanism, Neural Networks*

## I. INTRODUCTION

Remote sensing imagery plays a pivotal role in a wide array of applications, including environmental monitoring, urban planning, disaster management, and agricultural assessment. These applications demand accurate and efficient object detection methods to extract meaningful information from complex and large-scale images. However, detecting objects in remote sensing images presents unique challenges compared to natural image datasets. The diverse scales, orientations, and dense distribution of objects within cluttered backgrounds significantly hinder the performance of traditional detection algorithms.

Conventional object detection models, such as YOLO [1] and Faster R-CNN [2], have achieved significant success in natural images. However, these models often encounter limitations when applied to remote sensing imagery. Specifically, traditional methods struggle to manage objects with varying scales, accurately detect objects with arbitrary orientations, and adapt to the complex background clutter typical in remote sensing environments. Additionally, most conventional models rely on axis-aligned bounding boxes, which are not well-suited for the detection of oriented objects, leading to reduced accuracy. Several approaches have been proposed to address these issues, such as incorporating rotated bounding boxes and employing multi-scale feature extraction techniques. However, these methods often involve trade-offs between computational efficiency and detection accuracy, and may still fall short in environments with highly varied object scales and orientations. Moreover, these models' integration of attention mechanisms has been relatively limited, often focusing on spatial or channel attention separately, rather than leveraging their combined potential.

To overcome these challenges, we propose the Spatial Channel Attention Fusion Module (SCAF-Module). This module integrates multi-scale convolutions, adaptive rotated convolutions, and parallel spatial channel attention mechanisms to enhance detection accuracy. The multi-scale convolutions include a $3 \times 3$ rotated convolution, a $3 \times 3$ dilated rotated convolution, and a $5 \times 5$ dilated convolution, each contributing to the detection of objects at various scales and orientations. The spatial and channel attention mechanisms further refine these features, allowing the model to selectively focus on important regions and adapt to different object characteristics.

We define multi-scale convolution as the application of convolutional layers with different receptive fields, designed to capture features at various scales [3]. Adaptive rotated convolution [4] is defined as a type of convolution that adapts to the orientation of objects, enhancing the model's sensitivity to rotated objects. The Spatial Channel Attention Fusion is a technique that enhances feature representation by focusing on significant spatial regions and feature channels [5].

Our contributions are as follows:

- Multi-scale and Adaptive Rotated Convolutions: By incorporating convolutions with different receptive fields and adaptive rotated convolutions, the SCAF-Module effectively captures objects of varying scales and orientations.
- Spatial and Channel Attention Mechanisms: These mechanisms enhance the model's ability to focus on significant regions and channels, improving detection performance.
- Comprehensive Evaluation: Extensive experiments on the DOTA-v1.0 and HRSC2016 datasets demonstrate the SCAF-Module's effectiveness, achieving mAP scores of 80.94% and 98.23%, respectively.

This work was supported by the National Natural Science Foundation of China under Grant No. 61702340.

The remainder of this paper is organized as follows: Section II reviews related work in remote sensing object detection. Section III details the proposed SCAF-Module and its integration into the backbone network. Section IV presents the experiment setup, results, and ablation studies. Finally, Section TABLE VI. concludes the paper and discusses future work.

## II. RELATED WORK

### A. Remote Sensing Object Detection Frameworks

Object detection in remote sensing images has garnered significant attention due to its critical applications in areas such as urban planning, disaster management, and environmental monitoring. Traditional object detection frameworks designed for natural images, such as YOLO and Faster R-CNN, have been adapted to remote sensing scenarios. However, these frameworks face challenges unique to remote sensing images, such as the need to detect objects at various orientations and scales. Consequently, specialized frameworks have been developed to address these challenges, focusing on rotated object detection, multi-scale feature extraction, and robust performance in diverse and cluttered environments.

To mitigate the abundance of rotated anchors and to minimize the disparity between the feature representations and the actual objects, Ding et al. [6] have introduced the RoI transformer. This technique, which extracts rotated RoIs from the horizontal ones yielded by the RPN, substantially enhances the precision of detecting objects with orientation. Nonetheless, the incorporation of fully-connected layers and the RoI alignment process during the learning phase adds a layer of complexity and computational demands to the network.

To tackle the detection of small-scale, densely packed, and rotated objects, Yang et al. [7] have crafted an oriented object detection approach that integrates with the established Faster R-CNN framework. Additionally, a novel representation for oriented objects, known as gliding vertexes [8], has been put forward. This method refines the detection process by acquiring four vertex gliding offsets from the regression component of the Faster R-CNN architecture.

Despite these advancements, the reliance on horizontal RoIs for classification and oriented bounding box regression in these methods leads to significant misalignment issues between the objects and their corresponding features. Furthermore, various studies have delved into one-stage or anchor-free oriented object detection frameworks, which forgo the need for region proposal generation and RoI alignment, directly outputting object classes and oriented bounding boxes. For instance, a refined one-stage oriented object detector [9] has been proposed, featuring two pivotal enhancements: feature refinement and progressive regression, addressing the misalignment of features. A new label assignment strategy for one-stage oriented object detection, inspired by RetinaNet, dynamically assigns anchors as either positive or negative through an innovative matching approach. A single-shot alignment network (S$^2$ANet) [10] has been introduced for oriented object detection, focusing on harmonizing the classification score with location precision through deep feature alignment. Lastly, a dynamic refinement network (DRN) [11]has been conceptualized for oriented object detection, leveraging the anchor-free detection approach of CenterNet [12].

### B. Addressing Metric and Loss Inconsistency

The issue of inconsistency between metrics and loss functions is prevalent in horizontal bounding box object detection and becomes even more pronounced in remote sensing object detection due to the introduction of angle parameters. In horizontal bounding box detection, new IoU (Intersection over Union) calculation methods like DIoU (Distance Intersection over Union) [13] and GIoU (Generalized Intersection over Union) [14] have been proposed to alleviate inconsistency problems. However, these methods are non-differentiable and thus not directly applicable to remote sensing object detection.

Existing solutions to inconsistency in remote sensing object detection are limited, primarily focusing on designing new loss functions. These can be categorized into bounding box-based, pixel-based, and Gaussian distribution-based loss functions. Most current detection methods calculate the IoU of two inclined bounding boxes, often using smooth L1 as the regression loss function. However, for near-square targets, high IoU can still result in a significant loss. To address this, Yang et al. [7] proposed the IoU-smooth L1 loss, which combines IoU and smooth L1 to mitigate the problem. The overlapping forms of two inclined bounding boxes vary greatly. Zheng et al. [15] addressed this by proposing a rotation-robust IoU (RIoU) calculation method for 3D object detection, which can also be applied to 2D rotated object detection. This method defines a pair of projected rectangles to calculate the overlap area, allowing for the regression of bounding boxes at any angle. For anchor-free detection methods, Guo et al. [16] proposed using a convex hull formed by a set of irregular points to represent each rotated target, then optimizing the detector using a convex hull-based CIoU (Complete Intersection over Union) loss. Additionally, the smooth L1 loss is insensitive to large aspect ratio targets. To address this, Chen et al. [17] proposed PIoU (Pixels Intersection over Union) loss, which determines whether pixels are within the rotated box and calculates the rotated IoU by accumulating these pixels. This loss function can be applied to both anchor-based and anchor-free frameworks, though its accuracy needs improvement.

### C. Gaussian-based Loss

Recently, methods based on 2D Gaussian distributions have garnered significant attention. Yang et al. [18] analyzed the impact of angle differences, center point deviations, and different aspect ratios between rotated candidate boxes and ground truth on loss function changes. They designed a new loss function based on Gaussian Wasserstein distance. The approach involves converting rotated bounding boxes to 2D Gaussian distributions, calculating the Gaussian Wasserstein distance between the distributions of the ground truth and predicted boxes to derive the new loss function. However, this method lacks scale invariance, and optimizing only the rotation center can lead to positional deviations in the detection results. To address the scale variation issue brought by Gaussian Wasserstein distance, KL divergence [19] has been used as a substitute for loss calculation. This method, similar to Gaussian Wasserstein distance, derives a theoretical explanation for selecting distribution distance metrics to maintain detection

accuracy and scale invariance after transforming parameters into 2D Gaussian distributions. Both loss functions introduce additional hyperparameters, but the key to maintaining consistency between evaluation and loss is ensuring their trends remain consistent. Inspired by Kalman filtering, the KFIoU (Kalman Filter Intersection over Union) loss [20] was proposed. The basic steps involve modeling the ground truth and predicted bounding boxes as Gaussian distributions, aligning the center points of the two distributions, obtaining the Gaussian distribution of the overlapping area through Kalman filtering, and converting it back to a rotated bounding box. This approach approximates the rotated IoU.

### D. Backbone Network Design

Designing an effective backbone network for remote sensing images is crucial due to the varying scales and orientations of objects. Li et al. [3] introduced a strategy incorporating large kernel convolutions with different receptive fields into the backbone network. This approach dynamically adjusts the receptive fields to capture features at various scales. However, large kernel convolutions may lead to information loss for small objects, as their large receptive fields might cover multiple objects or noise regions. Pu et al. [4] employed adaptive rotated convolutions, rotating the convolutional kernels to achieve rotational sampling. While effective, rotating large kernels significantly increases computational complexity without a corresponding improvement in accuracy. This trade-off highlights the need for a balanced approach that can capture features at different scales and orientations without excessive computational cost.

### E. Attention Mechanisms

Attention mechanisms serve as a straightforward yet potent means of augmenting neural network representations across a multitude of applications. The channel-wise attention mechanism, exemplified by the SE block [5], leverages the insights from global averaging to recalibrate the importance of feature channels. Concurrently, spatial attention schemes such as those found in GENet [21], GCNet [22], and SGE [23], fortify the network's capacity to incorporate contextual cues through spatial filtering techniques. The CBAM [24] and BAM [25] architectures amalgamate channel and spatial attention, capitalizing on the strengths of both to refine feature representation.

Our proposed method focuses on the integration of backbone network design and attention mechanisms. Li et al. [3] incorporated prior knowledge from remote sensing images to develop large kernel selective convolutions; however, these large kernel convolutions can reduce the network's sensitivity to small objects, which are prevalent in remote sensing imagery. Pu et al. [4] designed rotated convolutions that are sensitive to object angles, yet this approach introduces significant computational overhead. In contrast, our work combines both strategies by implementing spatial attention, utilizing smaller rotated convolutions alongside larger standard convolutions. This approach effectively balances the detection of small and large objects without substantially increasing computational costs. Additionally, channel attention mechanisms [5] are employed to suppress irrelevant features while enhancing the

importance of relevant ones, thereby improving overall detection performance.

## III. PROPOSED METHOD

In this section, we introduce the architecture and components of the Spatial Channel Attention Fusion Module (SCAF-Module), designed to enhance remote sensing object detection by integrating multi-scale convolutions, adaptive rotated convolutions, and parallel spatial channel attention mechanisms. We also detail the overall backbone structure, which incorporates the SCAF-Module into a hierarchical framework to effectively capture and represent features at multiple levels. The SCAF-Module is built upon the following assumptions: (1) Objects in remote sensing images vary greatly in scale and orientation [20], necessitating a detection method that can handle these variations effectively. (2) Multi-scale [3] and adaptive rotated convolutions [4] are effective in capturing detailed features across different scales and orientations. (3) Spatial and channel attention mechanisms [5] can further enhance the detection performance by emphasizing important features.

### A. Spatial Channel Attention Fusion Module

The Spatial Channel Attention Fusion Module (SCAF-Module) is designed to enhance feature representation by integrating spatial [3] and channel attention [5] mechanisms with multi-scale and rotated convolutions [4]. This module aims to address the challenges posed by the diverse scales and orientations of objects in remote sensing images. The overall structure of the SCAF-Module is depicted in Fig. 1.

#### 1) Multi-scale Convolutions

The SCAF-Module begins by processing the input feature map $X$ through three convolutional layers, each with a different receptive field to capture features at various scales. The three convolutional layers include:

- $3 \times 3$ Rotated Convolution: Captures fine-grained details and small-scale features with enhanced sensitivity to object orientations, improving detection accuracy for rotated objects.
- $3 \times 3$ Rotated Dilated Convolution (dilation rate = 2): Utilizes a dilation rate of 2 to expand the receptive field without increasing the number of parameters, capturing medium-scale features while maintaining orientation adaptability.
- $5 \times 5$ Standard Dilated Convolution (dilation rate = 2): Further expands the receptive field to capture larger-scale features, ensuring comprehensive feature extraction across various object scales.

These layers generate three feature maps $F_1$, $F_2$, and $F_3$ respectively. The use of different receptive fields allows the module to balance sensitivity to both large and small objects, which is crucial for the diverse object sizes found in remote sensing images.

#### 2) Rotated Convolution

Rotated convolution is designed to address the unique challenges posed by remote sensing images, particularly the diverse orientations of objects. Traditional convolutional layers are limited by their fixed orientation, which can hinder the model's ability to accurately capture features of rotated objects. The rotated convolution mechanism introduces a way to

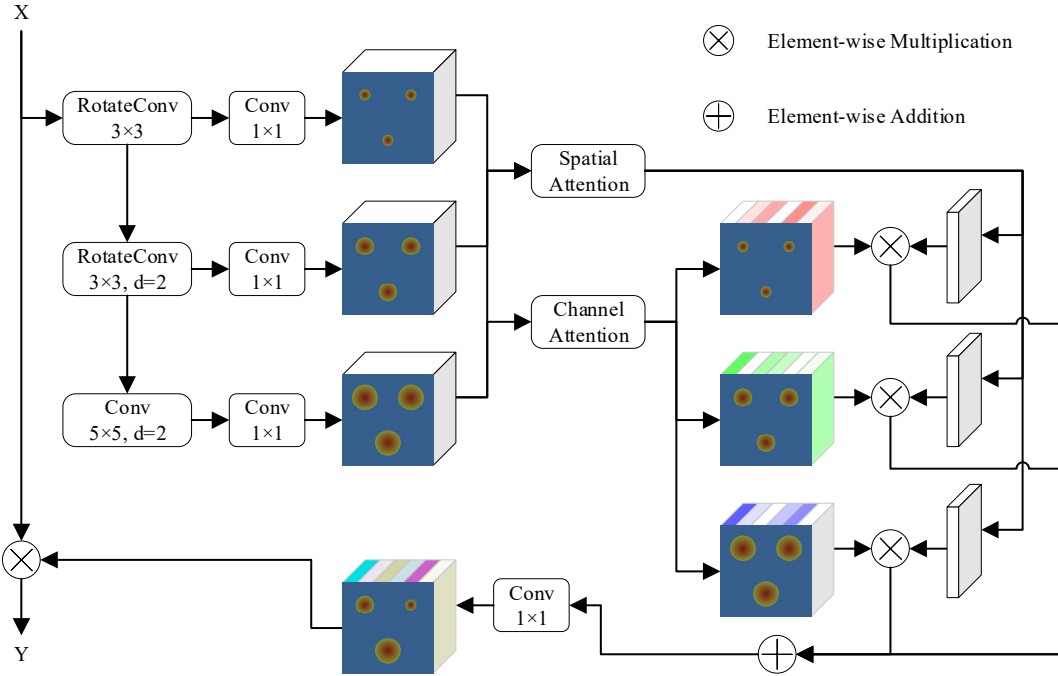

Fig. 1. The overall structure of the SCAF-Module.

dynamically adjust the orientation of the convolutional filters, enabling better alignment with the objects in the input image. The primary advantage of rotated convolution is its ability to rotate the convolutional kernels to match the orientation of the target objects. This is particularly beneficial for remote sensing applications where objects such as buildings, vehicles, and agricultural fields can appear at various angles. By aligning the convolutional filters with the orientation of these objects, the rotated convolution can more effectively capture the relevant features, leading to improved detection accuracy. The process of rotated convolution involves the following steps:

• Kernel Rotation: The convolutional kernel is treated as a set of sampling points in the kernel space. These sampling points are then rotated by an angle θ, which is dynamically determined based on the input feature map. This rotation allows the kernel to align with the orientation of the objects in the image.

• Bilinear Interpolation: After rotating the sampling points, bilinear interpolation is used to map the original convolution parameters to the new rotated positions. This ensures that the rotated kernel retains the characteristics of the original kernel while adapting to the new orientation.

• Dynamic Angle Generation: The rotation angle $\theta$ is not fixed but is generated dynamically by a routing function based on the input features. This allows the model to adapt to different orientations present in the input image, providing a flexible and robust solution for capturing rotated objects.

Rotated convolution addresses the limitations of traditional convolutional layers by introducing orientation adaptability. This innovation is crucial for improving the accuracy of object detection in remote sensing images, where the orientation of objects is often varied and unpredictable. By aligning the convolutional filters with the objects' orientations, the SCAF-

Module can more effectively capture and represent the relevant features, leading to superior detection performance.

*3) Spatial Attention Mechanism*

The spatial attention mechanism is designed to focus on important regions within the feature maps, addressing the variability in the shapes and scales of objects. As shown in Fig. 2, the spatial attention mechanism operates as follows:

• Concatenation: The feature maps $F_1$, $F_2$, and $F_3$ are concatenated along the channel dimension to form a combined feature map $F$.

• Pooling: The combined feature map $F$ undergoes average pooling and max pooling along the channel dimension, producing the average feature map $AF$ and maximum feature map $MF$.

• Fusion: The pooled feature maps $AF$ and $MF$ are concatenated along the channel dimension to form the fused feature map $AMF$.

• Convolution and Activation The fused feature map $AMF$ is passed through a convolutional layer and a sigmoid activation function to produce the spatial attention map $SF$:

$$SF = \sigma(\text{Conv}([\text{AvgPool}(F), \text{MaxPool}(F)])), \qquad (1)$$

where σ denotes the sigmoid activation function, which maps the output to a range of [0, 1], serving as the spatial attention weights. This mechanism enables the model to selectively focus on significant regions in the feature maps, enhancing the detection of objects with varying shapes and scales.

*4) Channel Attention Mechanism*

The channel attention mechanism dynamically adjusts the weights of different channels, emphasizing channels that are more informative for the task at hand. This mechanism is crucial

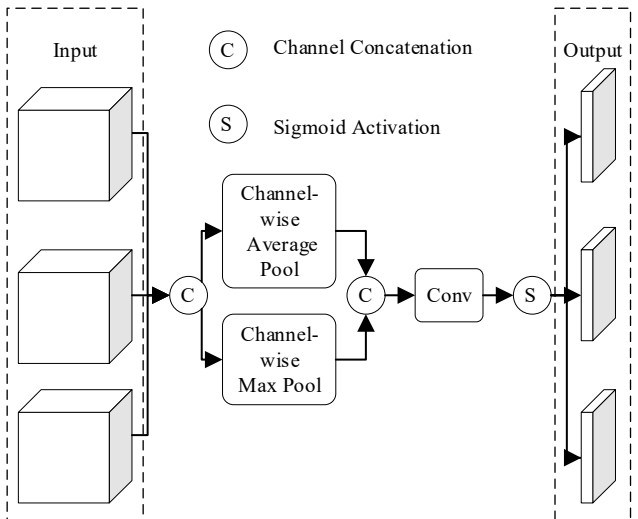

Fig. 2. Spatial attention mechanism.

for optimizing the feature representation by selectively enhancing the most relevant channels based on the global context of the feature map. As shown in Fig. 3, the channel attention mechanism operates as follows:

- Global Average Pooling: Each feature map $F_i$ undergoes global average pooling to capture the global context of the feature map, resulting in a descriptor vector $z$. The global average pooling operation aggregates the spatial information across the entire feature map, producing a channel-wise descriptor that summarizes the global context.
- Squeeze and Excitation: The descriptor vector $z$ is passed through two convolutional layers. The first convolutional layer reduces the number of channels by a ratio (typically 16), and the second convolutional layer restores the original number of channels. This sequence of operations is designed to learn the importance of each channel dynamically:

$$s_c = \sigma(\text{Conv}_2(\text{ReLU}(\text{Conv}_1(z)))), \qquad (2)$$

where $\text{Conv}_1$ and $\text{Conv}_2$ are the convolutional layers used for compression and excitation, respectively, and $\sigma$ denotes the sigmoid activation function.

- Reweighting: The learned channel weights $s_c$ are applied to the corresponding feature map $F_i$, producing the channel-weighted feature map $CF_i$:

$$CF_i = F_i \odot s_c, \qquad (3)$$

where $\odot$ denotes the element-wise multiplication.

The core idea behind this mechanism is to use global information to recalibrate the feature map in a channel-wise manner, enhancing the model's ability to focus on the most informative channels and improving the overall feature representation. By integrating this SE-based channel attention mechanism, the SCAF-Module can effectively capture and utilize the global context, leading to improved performance in remote sensing object detection tasks.

*5) Feature Fusion*
The outputs from the spatial and channel attention mechanisms are element-wise multiplied and summed to produce a new feature map $FF$:

$$FF = \sum_{i=1}^{3} S F_i \odot CF_i. \qquad (4)$$

where $\odot$ denotes the element-wise multiplication. This fusion step integrates spatial and channel attention, enhancing the feature representation by focusing on both important regions and informative channels. Finally, the fused feature map $FF$ is element-wise multiplied with the input feature map $X$ to produce the final output $Y$ of the module:

$$Y = X \odot FF. \qquad (5)$$

This final step ensures that the enhanced features are integrated with the original input, maintaining the integrity of the input information while incorporating the attention mechanisms' enhancements. The SCAF-Module, through its combination of multi-scale convolutions, spatial attention, and channel attention, effectively addresses the challenges of detecting objects with varying scales and orientations in remote sensing images.

*B. Backbone Structure*

The backbone structure of the SCAF-Module is meticulously designed to effectively capture and process the diverse features present in remote sensing images. This section provides an in-depth look at the backbone architecture, consisting of multiple stages, each composed of several blocks that integrate the SCAF-Module to enhance feature representation.

*1) Block Structure*
Each block within the backbone is constructed to maintain the shape and channel dimensions of the input while enhancing the feature representation through a series of operations. The block structure is as follows:

- Normalization 1: The input feature map undergoes a normalization process to stabilize and accelerate the training process.
- Fully Connected Layer: A fully connected (FC) layer is applied to the normalized features, transforming them into a different feature space.

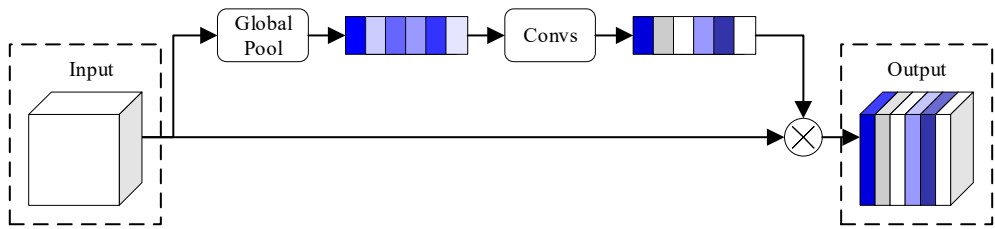

Fig. 3. Channel attention mechanism.

• GELU Activation: The output from the fully connected layer is passed through a GELU (Gaussian Error Linear Unit) activation function to introduce non-linearity.

• SCAF-Module: The activated features are then processed by the SCAF-Module, which applies multi-scale convolutions, spatial attention, and channel attention to enhance the feature representation.

• Fully Connected Layer: Another fully connected layer is applied to the features output by the SCAF-Module.

• Normalization 2: A second normalization layer is used to further stabilize the feature representations.

• MLP: Finally, the features pass through a multi-layer perceptron (MLP) for additional transformation and refinement.

The block incorporates two residual connections to preserve the original input features and prevent the degradation problem commonly encountered in deep networks. The first residual connection adds the input to the feature map before the second normalization step, while the second residual connection adds the input to the final output of the block, ensuring that the input-output shape and channel dimensions remain unchanged. The block structure is illustrated in Fig. 4.

### 2) Stage Structure
The backbone is organized into multiple stages, each consisting of several blocks to progressively extract and refine features at different scales and resolutions. Each stage operates as follows:

• Shape and Channel Adjustment: At the beginning of each stage, a convolutional layer adjusts the shape and channel dimensions of the input feature map to prepare it for further processing.

• Repeated Blocks: The adjusted feature map is then passed through a series of blocks. Each block applies the operations described above, progressively enhancing the feature representation.

• Normalization: The output of the final block in each stage undergoes normalization to ensure stable feature distribution before passing to the next stage.

The multi-stage structure allows the backbone to capture features at varying levels of abstraction, from low-level edges and textures to high-level semantic information.

### 3) Integration with Oriented R-CNN
The backbone is integrated into the Oriented R-CNN framework, replacing the original ResNet backbone. The Oriented R-CNN is specifically designed for object detection in remote sensing images, where objects often appear in arbitrary orientations. By incorporating the SCAF-Module-based backbone, the Oriented R-CNN benefits from improved feature extraction capabilities, particularly in handling the diverse scales and orientations of objects in remote sensing imagery.

The backbone structure leveraging the SCAF-Module significantly enhances the capability of the Oriented R-CNN to accurately detect and classify objects in remote sensing images. The multi-scale convolutions, combined with spatial and channel attention mechanisms, ensure that the model captures a comprehensive set of features, leading to superior detection performance.

## IV. Experiments

In this section, we present the experiments conducted to evaluate the performance of the proposed SCAF-Module. We detail the datasets used, the evaluation metrics, and the experiment setup. Finally, we present and analyze the results, demonstrating the effectiveness of our approach.

### A. Datasets and Evaluation
We evaluate the SCAF-Module on two widely used remote sensing image datasets: DOTA-v1.0 and HRSC2016. These datasets are chosen for their diversity in object scales, orientations, and complexity of scenes, which pose significant challenges for object detection models.

DOTA-v1.0 [26]: The following fifteen object classes are covered in this dataset: Plane (PL), Baseball diamond (BD), Bridge (BR), Ground track field (GTF), Small vehicle (SV), Large vehicle (LV), Ship (SH), Tennis court (TC), Basketball court (BC), Storage tank (ST), Soccer-ball field (SBF), Roundabout (RA), Harbor (HA), Swimming pool (SP), and Helicopter (HC). The dataset contains a wide variety of object scales and orientations, making it a suitable benchmark for evaluating our model's ability to handle diverse object characteristics. Due to the large size of images, offline data augmentation is typically used. For single-scale training and testing, images are cropped to 1024×1024 patches with 200 pixels overlap. For multi-scale training and testing, images are first resized to 0.5, 1.0, and 1.5 times their original size, and then cropped to 1024×1024 patches with 500 pixels overlap.

HRSC2016 [27]: This dataset focuses on ship detection and includes images captured from various angles and distances, with ships annotated using oriented bounding boxes. The variability in ship sizes and orientations makes this dataset an excellent testbed for our model's robustness.

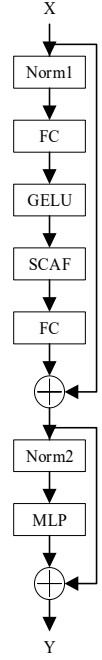

Fig. 4. Block structure.

The primary evaluation metric used in The experiments is the mean Average Precision (mAP), which measures the precision-recall performance across different object categories. We report the mAP scores to provide a comprehensive assessment of our model's detection capabilities.

## B. Experiment Setup

The experiments are conducted using the MMRotate framework on an NVIDIA GeForce RTX 3090 GPU with a batch size of 2 for training and evaluation. The optimizer used is AdamW with a learning rate of $5 \times 10^{-5}$, $\beta_1 = 0.9$, $\beta_2 = 0.999$, and a weight decay of 0.05. The learning rate follows a step policy with an initial linear warmup for 500 iterations starting at a third of the base learning rate, then decaying at epochs 8 and 11. Image normalization is applied with mean values [123.675, 116.28, 103.53] and standard deviations [58.395, 57.12, 57.375]. The training pipeline includes resizing, random flipping (horizontal, vertical, diagonal), random rotation, normalization, padding, and data collection. The test pipeline involves multi-scale augmentation and normalization.

The SCAF-Module is integrated into the Oriented R-CNN framework, replacing the original ResNet backbone, to evaluate its performance in detecting objects with arbitrary orientations in remote sensing images. For the ablation experiments, the backbone is not pre-trained on ImageNet to enhance experimental efficiency. In contrast, for the comparative experiments, the backbone undergoes pre-training on ImageNet for 300 epochs before being fine-tuned on the DOTA-v1.0 and HRSC2016 datasets to achieve higher performance.

## C. Comparative Experiments

In this section, we evaluate the performance of the SCAF-Module against six advanced models, including the baseline Oriented R-CNN, using two widely adopted remote sensing image datasets: DOTA-v1.0 and HRSC2016. The comparison focuses on mean Average Precision (mAP) as the primary metric.

### 1) DOTA-v1.0 Dataset

The DOTA-v1.0 dataset is a standard benchmark for remote sensing object detection. We conducted experiments using both single-scale and multi-scale training and testing protocols to assess the robustness of our model.

For the single-scale evaluation, large images from the DOTA-v1.0 dataset were divided into 1024×1024 patches with a 200-pixel overlap. The results, summarized in Table I, show that the SCAF-Module achieved a significant mAP of 78.96%, outperforming all compared models. This demonstrates the module's ability to effectively capture fine-grained details and accurately detect objects at a fixed scale. In the multi-scale evaluation, images were rescaled to 0.5, 1.0, and 1.5 times their original sizes, with a 500-pixel overlap during patching. As shown in Table II, the SCAF-Module achieved an mAP of 80.94%, again surpassing the other models. This result highlights the module's robustness in adapting to various object scales, a critical requirement in remote sensing tasks.

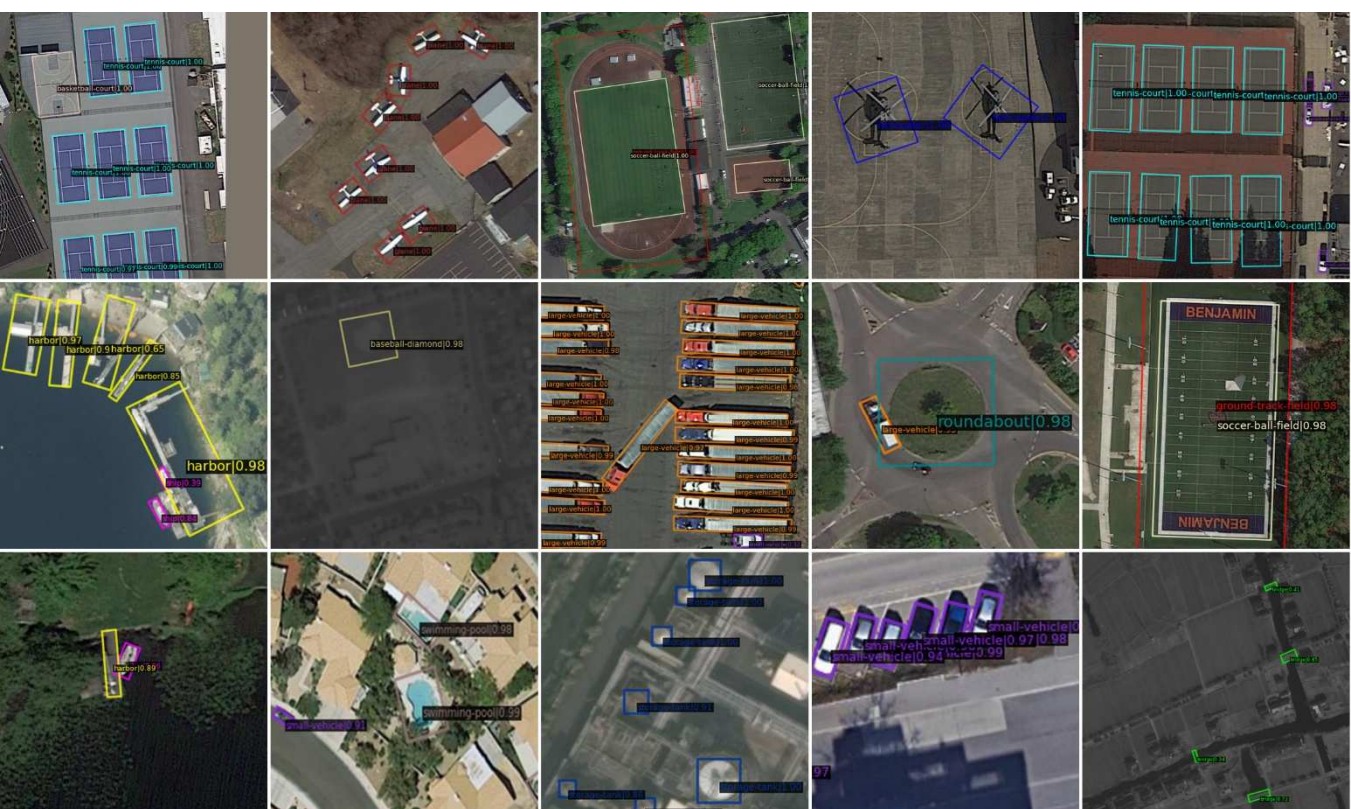

Fig. 5. Detection Results on DOTA-V1.0 Dataset.

These evaluations on the DOTA-v1.0 dataset confirm the superior performance of the SCAF-Module, particularly in its ability to enhance detection accuracy across both single and multi-scale scenarios. To visually represent the effectiveness of our model on the DOTA-v1.0 dataset, we present a series of detection result images in Fig. 5.

### 2) HRSC2016 Dataset

The HRSC2016 dataset focuses on ship detection, providing a rigorous test of model precision and robustness. We evaluated our model under the PASCAL VOC 2007 and VOC 2012 metrics to ensure a thorough assessment. As detailed in Table III, the SCAF-Module achieved mAP scores of 90.61% under the VOC 2007 metric and 98.23% under the VOC 2012 metric, marking a notable improvement over the other models. This performance can be attributed to the module's advanced attention mechanisms and multi-scale convolutional structure, which enhance its ability to detect ships with varying orientations—a frequent challenge in remote sensing imagery.

Overall, the results from the HRSC2016 dataset further validate the effectiveness of the SCAF-Module in detecting objects with diverse scales and orientations, reinforcing its value as a robust tool for remote sensing object detection tasks.

### D. Ablation Study

To understand the contribution of each component in the SCAF-Module, we conduct ablation studies on the DOTA-v1.0 dataset. The goal is to analyze the effects of Spatial Attention, Channel Attention, their order, and the use of Rotated Convolutions on the overall performance of the model.

Contribution of Individual Components

We investigate the contributions of various components to the overall performance of our model. The baseline configuration employs multi-scale convolutional layers without any additional mechanisms. We then incrementally add the following components: rotated convolutions, channel attention, and spatial attention. All experiments are conducted using single-scale training and testing on the DOTA-v1.0 dataset.

The baseline is multi-scale convolutional layers without any attention mechanisms. This setup serves as the foundational model, capturing features at different scales. We first incorporate spatial attention into the baseline. The spatial attention mechanism enables the model to focus on important spatial regions, enhancing the detection of objects of varying shapes and scales within the image. Next, we add the channel attention mechanism to the model with rotated convolutions. This mechanism allows the model to emphasize important channels, improving the representation of semantic information critical for accurate object detection. Finally, we integrate rotated convolutions into the model. Rotated convolutions enhance the model's ability to capture features at various orientations, crucial for remote sensing images where objects often appear in arbitrary orientations.

The results of these experiments are summarized in Table IV. Each row in the table shows the mAP achieved by the model with the addition of the respective component. The performance improvements with each added component demonstrate the effectiveness of both the rotated convolutions and the attention mechanisms in enhancing the detection capabilities of the model.

### 1) Order of Spatial and Channel Attention

We also tested different sequences of applying Spatial and Channel Attention. The results of these experiments are presented in Table V.

When comparing the sequences of applying Spatial and Channel Attention, it is observed that the parallel application of both attentions yields the best results. Sequential applications result in slightly lower mAP scores, indicating that the simultaneous focus on both spatial and channel aspects is more beneficial.

### 2) Effects of Rotated Convolutions

Finally, we tested the impact of using Rotated Convolutions in different configurations. The results of these experiments are presented in Table VI.

TABLE I. AP FOR EACH CATEGORY AND OVERALL MAP ON DOTA-V1.0 (SINGLE-SCALE).

| Model | PL | BD | BR | GTF | SV | LV | SH | TC | BC | ST | SBF | RA | HA | SP | HC | mAP |
|---|---|---|---|---|---|---|---|---|---|---|---|---|---|---|---|---|
| GWD [18] | 88.92 | 77.08 | 45.91 | 69.30 | 72.52 | 64.05 | 76.33 | 90.87 | 79.18 | 80.45 | 57.67 | 64.36 | 63.60 | 64.75 | 48.24 | 69.55 |
| R³Det [9] | 89.30 | 73.36 | 45.10 | 71.21 | 76.51 | 74.01 | 81.03 | 90.89 | 79.01 | 83.54 | 59.37 | 63.47 | 63.04 | 65.93 | 37.02 | 70.19 |
| S²A-Net [10] | 88.70 | 81.41 | 54.28 | 69.75 | 78.04 | 78.23 | 80.54 | 90.69 | 84.75 | 86.22 | 65.03 | 65.81 | 76.16 | 73.37 | 58.86 | 76.11 |
| ReDet [28] | 88.79 | 82.64 | 53.97 | 74.00 | 78.13 | 84.06 | 88.04 | 90.89 | 87.78 | 85.75 | 61.76 | 60.39 | 75.96 | 68.07 | 63.59 | 76.25 |
| Oriented R-CNN [29] | 88.86 | 83.48 | 55.27 | 76.92 | 74.27 | 82.10 | 87.52 | 90.90 | 85.56 | 85.33 | 65.51 | 66.82 | 74.36 | 70.15 | 57.28 | 76.28 |
| LSKNet [3] | 89.78 | 81.24 | 54.09 | 75.96 | 79.31 | 85.13 | 88.49 | 90.90 | 87.41 | 84.87 | 64.12 | 64.31 | 77.03 | 78.22 | 67.02 | 77.86 |
| SCAF(ours) | 89.72 | 85.25 | 55.38 | 76.10 | 79.55 | 84.85 | 88.43 | 90.85 | 87.46 | 85.71 | 66.99 | 68.54 | 76.85 | 79.79 | 68.87 | **78.96** |

TABLE II. MAP ON DOTA-V1.0 (MULTI-SCALE).

| Model | mAP |
|---|---|
| R³Det | 76.47 |
| S²A-Net | 79.42 |
| ReDet | 79.87 |
| GWD | 80.23 |
| LSKNet | 80.32 |
| O-RCNN | 80.62 |
| SCAF(ours) | **80.94** |

TABLE III. MAP ON HRSC2016 (VOC 2007 AND VOC 2012).

| Model | mAP(07) | mAP(12) |
|---|---|---|
| S²A-Net | 90.17 | 95.01 |
| R3Det | 89.26 | 96.01 |
| GWD | 89.85 | 97.37 |
| O-RCNN | 90.50 | 97.60 |
| ReDet | 90.46 | 97.63 |
| LSKNet | 90.27 | 97.80 |

| | | |
|---|---|---|
| SCAF(ours) | **90.61** | **98.23** |

TABLE IV.    CONTRIBUTION OF INDIVIDUAL COMPONENTS.

| Configuration | mAP |
|---|---|
| Baseline | 67.62 |
| + Spatial Attention | 68.45 |
| + Channel Attention | 69.32 |
| + Rotated Convolution | **69.79** |

TABLE V.    ORDER OF SPATIAL AND CHANNEL ATTENTION.

| Configuration | mAP |
|---|---|
| Spatial then Channel Attention | 67.77 |
| Channel then Spatial Attention | 68.44 |
| Parallel Spatial & Channel Attention | **69.32** |

TABLE VI.    EFFECTS OF ROTATED CONVOLUTIONS.

| Configuration | mAP |
|---|---|
| All ordinary convolutions | 69.32 |
| The first convolution rotated | 69.59 |
| The first and second convolutions rotated | **69.79** |
| All convolutions rotated | 69.70 |

In the experiments focusing on the Rotated Convolutions, replacing the first and second convolutions with rotated ones gives the best performance. Using only ordinary convolutions or replacing all convolutions with rotated ones results in lower mAP scores. This suggests that a balanced combination of ordinary and rotated convolutions is most effective for capturing diverse object orientations in remote sensing images.

The ablation study demonstrates the effectiveness of each component within the SCAF-Module. Spatial Attention enhances the model's ability to focus on important regions within the image, Channel Attention dynamically adjusts the significance of different feature channels, and Rotated Convolutions improve the detection of objects with varying orientations. The combination of these components results in a significant performance boost, validating the design choices made in the development of the SCAF-Module.

## V. CONCLUSION

In this paper, we introduced the Spatial Channel Attention Fusion Module (SCAF-Module), a novel approach designed to enhance remote sensing object detection. By integrating multi-scale convolutions, adaptive rotated convolutions, and parallel spatial channel attention mechanisms, our model effectively addresses the challenges posed by the diverse scales and orientations of objects in remote sensing images.

The experiment results on the DOTA-v1.0 and HRSC2016 datasets demonstrate the effectiveness of the SCAF-Module. Specifically, the module achieved impressive mean Average Precision (mAP) scores of 80.94% and 98.23% on the DOTA-v1.0 and HRSC2016 datasets, respectively. These results underscore the adaptability and robustness of our approach in handling various object detection scenarios in remote sensing imagery. Furthermore, the ablation studies validate the individual contributions of the spatial and channel attention mechanisms, as well as the impact of rotated convolutions on improving detection accuracy. The comparative experiments show that the SCAF-Module outperforms several advanced models, including the baseline Oriented R-CNN, highlighting its superior performance.

Overall, the SCAF-Module offers a significant advancement in remote sensing object detection by providing a more comprehensive and adaptable framework. Future work will focus on further optimizing the module and exploring its application in other challenging remote sensing tasks.

### ACKNOWLEDGMENT

This work was supported by the National Natural Science Foundation of China under Grant No. 61702340.

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
