# OpenReview forum: "Remote Sensing Object Detection Based on Fusion of Spatial and Channel Attention"
_IEEE.org/ICIST/2024/Conference — IEEE ICIST 2024 Conference Submission_

### Official Review · Reviewer_3TWW · 2024-08-21
**Accept**

**Rating:** 7
**Confidence:** 3

**Review:**

This paper introduced the SCAF-Module, a novel approach designed to enhance remote sensing object detection. The theory is correct and can be accepted after responding the following comments.
(1)	What is the contribution of the paper? It should be highlighted both in the introduction and in the content.
(2)	It can be compared with existing articles to make the innovation point clearer.
(3)	please check if you need to update your Introduction/Related Work section to include latest closely relevant references that have appeared in journals and/or conferences in the past two years.

---

### Official Review · Reviewer_HoPm · 2024-08-22
**This article is very interesting and a good one**

**Rating:** 7
**Confidence:** 3

**Review:**

This paper introduces the SCAF-Module, which is devised to elevate detection accuracy through the integration of multi-scale convolutions, adaptive rotated convolutions, and parallel spatial channel attention mechanisms. The obtained result is valuable and can be accepted if the following problems can be clarified.
 (1) In the introduction, the shortages of those relevant studies are suggested to be further summarized.
(2) There exist several spelling and grammar errors. Please check carefully and further polish
(3) In the Experiments section, the main results of this article can be appropriately simplified and explained.
 (4) The references require updating and standardization of their format to ensure consistency and accuracy.

---

### Official Review · Reviewer_i4av · 2024-08-26
**Remote Sensing Object Detection based on Fusion of Spatial and Channel Attention**

**Rating:** 7
**Confidence:** 2

**Review:**

This paper proposed the Spatial Channel Attention Fusion Module (SCAF-Module), designed to enhance detection accuracy by integrating multi-scale convolutions, adaptive rotated convolutions, and parallel spatial channel attention mechanisms. The obtained result is valuable and can be accepted if the following problems can be clarified.
1. There are a few language issues throughout the paper.
2. The paper should include comparisons against the existing literature to demonstrate its advantages.
3. The paper should be added to the Assumptions and definitions with relative references to show the rationality of this paper.

---

### Comment · Reviewer_HoPm · 2024-08-21
**This article is very interesting and a good one**

This paper introduces the SCAF-Module, which is devised to elevate detection accuracy through the integration of multi-scale convolutions, adaptive rotated convolutions, and parallel spatial channel attention mechanisms. The obtained result is valuable and can be accepted if the following problems can be clarified.
(1)	In the introduction, the shortages of those relevant studies are suggested to be further summarized.
(2)	There exist several spelling and grammar errors. Please check carefully and further polish
(3)	In the Experiments section, the main results of this article can be appropriately simplified and explained.
(4)	The references require updating and standardization of their format to ensure consistency and accuracy.

---

### Decision · Program_Chairs · 2024-09-06

Accept (Oral)